Brief Communication

# Ultrafast and accurate sequence alignment and clustering of viral genomes

Andrzej Zielezinski [1,6], Adam Gudyś [2,6], Jakub Barylski [3], Krzysztof Siminski[2], Piotr Rozwalak[1,4], Bas E. Dutilh [4,5] ✉ & Sebastian Deorowicz [2] ✉

Viromics produces millions of viral genomes and fragments annually, overwhelming traditional sequence comparison methods. Here we introduce Vclust, an approach that determines average nucleotide identity by Lempel–Ziv parsing and clusters viral genomes with thresholds endorsed by authoritative viral genomics and taxonomy consortia. Vclust demonstrates superior accuracy and efficiency compared to existing tools, clustering millions of genomes in a few hours on a mid-range workstation.

Metagenomics and viromics are identifying new viruses at an unprecedented rate, but recognizing which sequences were seen before remains challenging[1,2]. Calculating average nucleotide identity (ANI), essential for classification, is limited by the scalability of alignment tools like anicalc[3], commonly used to cluster viruses into virus operational taxonomic units (vOTUs), or VIRIDIC[4], recommended by the International Committee on Taxonomy of Viruses (ICTV) to delineate bacteriophage species and genera. Large-scale sequence comparisons rely on efficient, albeit less accurate, k-mer approaches such as sketching (FastANI[5]) or sparse approximate alignments (skani[6]). Moreover, most tools lack clustering functionality or do not scale to large metagenomic datasets (Extended Data Table 1).

Vclust is a fast alignment-based method that calculates ANI measures for complete and fragmented viral genomes and clusters them according to ICTV and Minimum Information about an Uncultivated Virus Genome (MIUViG) standards[1,4] (Extended Data Table 1). It introduces three components (Fig. 1a). First, Kmer-db 2, a successor of Kmer-db[7], rapidly determines related genomes using either all k-mers or a predefined fraction. Second, LZ-ANI, a Lempel–Ziv parsing-based algorithm (Fig. 1b and Methods), identifies local alignments within related genome pairs and calculates overall ANI from these aligned regions with high sensitivity and accuracy. Third, Clusty efficiently implements six clustering algorithms suited for sparse distance matrices with millions of genomes (Fig. 1c).

We first tested Vclust's accuracy of total average nucleotide identity (tANI) estimation (Fig. 1d) among 10,000 pairs of phage genomes

containing simulated mutations, including substitutions, deletions, insertions, inversions, duplications and translocations (Methods and Supplementary Table 1). Vclust and VIRIDIC, both alignment-based tools, provided tANI values close to the expected ones, with mean absolute error (MAE) values of 0.3% and 0.7%, respectively, outperforming FastANI (6.8%) and skani (21.2%; Fig. 2a). Vclust predictions consistently approached expected values as tANI increased, while VIRIDIC underestimated tANI (Fig. 2a). Among genome pairs above the ICTV's species threshold (tANI ≥ 95%[4], n = 1,188), Vclust reported only 22 pairs below the threshold, whereas VIRIDIC underestimated nearly 10× more (n = 210; Supplementary Table 2).

Next, we determined tANI using VIRIDIC in an all-to-all comparison of 4,244 bacteriophage genomes. Vclust had a higher correlation with VIRIDIC tANI (Pearson's r = 0.983) than skani (r = 0.902) and FastANI (r = 0.671) across the entire tANI range ≥ 70% (22,606 genome pairs; Fig. 2b) and outperformed both tools within their reliability range ≥ 80%[5,6].

Then, we compared the consistency of the bacteriophage species groupings (tANI ≥ 95%) with the official ICTV taxonomy (Methods). Vclust and VIRIDIC showed moderate agreement with ICTV (73% and 69%, respectively), followed by FastANI (40%) and skani (27%). Upon examining genome pairs where both Vclust and VIRIDIC diverged from the ICTV's classification, we found inconsistencies in 50 ICTV taxonomic proposals (Supplementary Tables 3 and 4). Excluding these cases improved the agreement of both tools with ICTV taxonomy, with Vclust retaining superiority (95%) over VIRIDIC (90%) and the other

[1]Department of Computational Biology, Faculty of Biology, Adam Mickiewicz University, Poznan, Poland. [2]Faculty of Automatic Control, Electronics and Computer Science, Silesian University of Technology, Gliwice, Poland. [3]Department of Molecular Virology, Faculty of Biology, Adam Mickiewicz University, Poznan, Poland. [4]Institute of Biodiversity, Faculty of Biological Sciences, Cluster of Excellence Balance of the Microverse, Friedrich Schiller University Jena, Jena, Germany. [5]Theoretical Biology and Bioinformatics, Science4Life, Utrecht University, Utrecht, the Netherlands. [6]These authors contributed equally: Andrzej Zielezinski, Adam Gudyś. ✉e-mail: b.e.dutilh@uni-jena.de; sebastian.deorowicz@polsl.pl

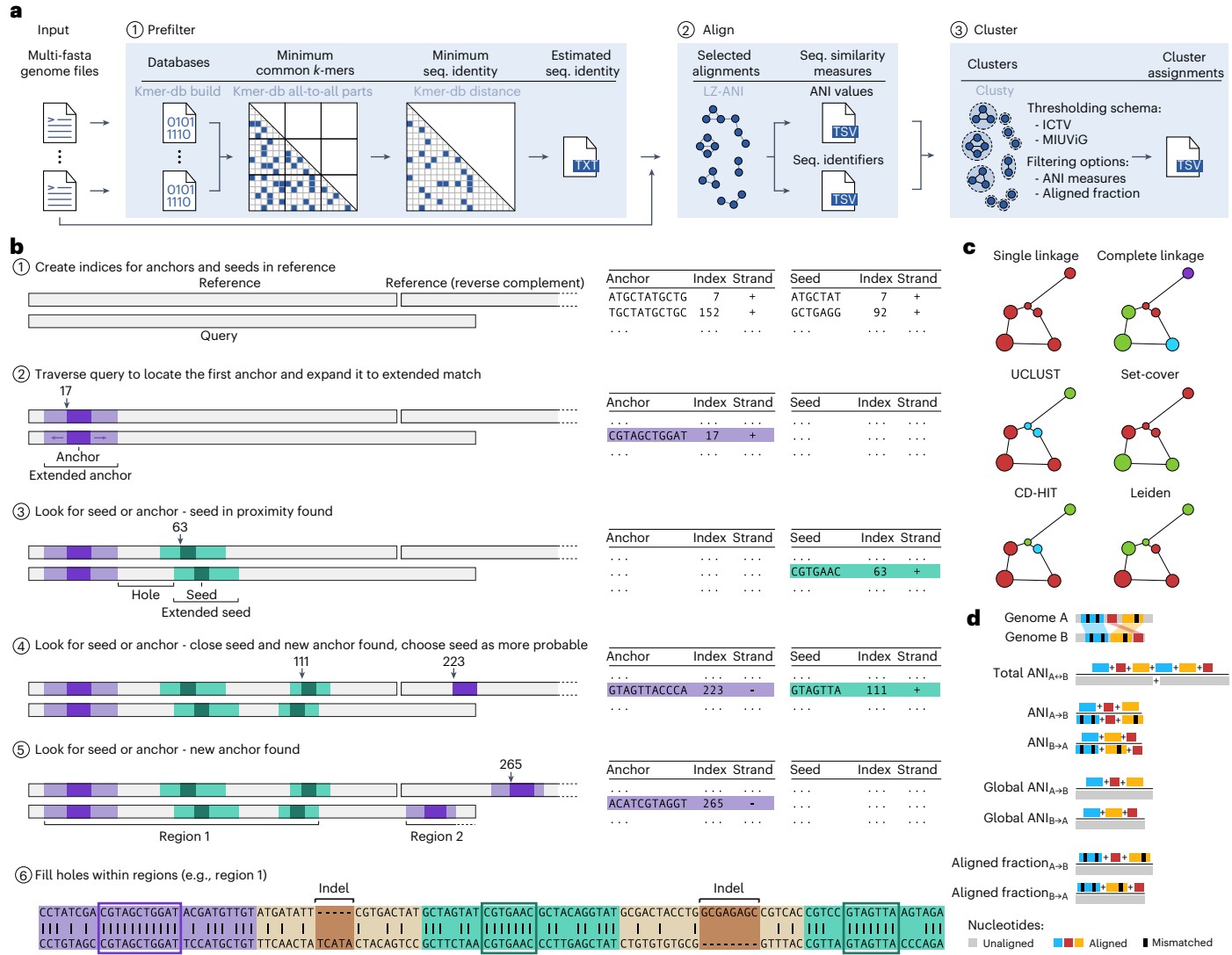

**Fig. 1 | Vclust algorithm and features. a**, Vclust's workflow: (1) prefilter similar genome sequence pairs with sufficient *k*-mer-based identity estimated using Kmer-db 2; (2) align similar genome pairs and calculate ANI using LZ-ANI; and (3) cluster genomes based on defined cutoffs using Clusty. **b**, Sequence alignment using Lempel–Ziv parsing (Methods). **c**, Vclust's clustering algorithms. Vertex size represents genome sequence length, and edge lengths indicate the distance (1 − ANI) between genomes. A more detailed depiction of the clustering algorithms is shown in Extended Data Fig. 1. **d**, Illustration of the calculation of Vclust's sequence similarity measures.

tools (Supplementary Table 5). For genus groupings (tANI ≥ 70%), Vclust achieves 92% agreement with ICTV taxonomy, comparable to VIRIDIC's 93%, despite inconsistent application of the threshold we found across ICTV genera (Supplementary Tables 6 and 7 and Extended Data Fig. 2). Given Vclust's high agreement with ICTV taxonomy, accurate tANI determination and processing speed >40,000× faster than VIRIDIC (Fig. 2c and Supplementary Table 8), it emerges as the prime tool for bacteriophage classification.

We then assessed Vclust's accuracy in matching contig pairs that satisfy MIUViG thresholds (ANI ≥ 95% and aligned fraction (AF) ≥ 85%; Fig. 1d). We subsampled over 90,000 metagenomic contigs from the IMG/VR database and used BLASTn[8]+ anicalc[3] (most accurate alignment-based method) to identify over 4 million sequence pairs that met MIUViG thresholds. Vclust recovered the highest number of pairs (99%), followed by MegaBLAST + anicalc (97%), skani (96%, or 86% in the fastest mode), FastANI (96%) and MMseqs2 (ref. 9) (70%; Fig. 2d and Supplementary Table 9). Both Vclust and Mega-BLAST produced ANI and AF estimates consistently with the BLASTn values (Pearson *r* > 0.96), outperforming the other tools (*r* = 0.2–0.8).

On average, ANI and AF values obtained by Vclust and MegaBLAST showed minimal deviation from the expected values (MAE < 1%; Supplementary Table 9), with Vclust having the narrowest error range among all the tools (Fig. 2d). This trend is consistent across varying contig sizes, from smallest (<5 kb) to largest (>100 kb; Supplementary Table 10).

The scalability of the tools was tested using the entire IMG/VR database of 15,677,623 virus contigs. Vclust performed sequence identity estimations for ~123 trillion contig pairs and alignments for ~800 million pairs, resulting in 5–8 million vOTUs depending on the clustering algorithm (Supplementary Table 11 and Supplementary Fig. 1). These vOTUs are generally consistent with those identified by MegaBLAST, with Vclust clustering approximately 75,000 more contigs on average, indicating higher sensitivity (Supplementary Table 11). Vclust was >115× faster than MegaBLAST, >6× faster than skani or FastANI, and ~1.5× faster than MMseqs2 (Fig. 2e,f, Extended Data Fig. 3 and Supplementary Table 12). Although skani in its fastest mode was 7× faster than Vclust (Supplementary Table 12), it was sub-stantially less accurate (Supplementary Table 9). In addition, Vclust's runtime and memory usage can be further reduced by ~40% and ~60%,

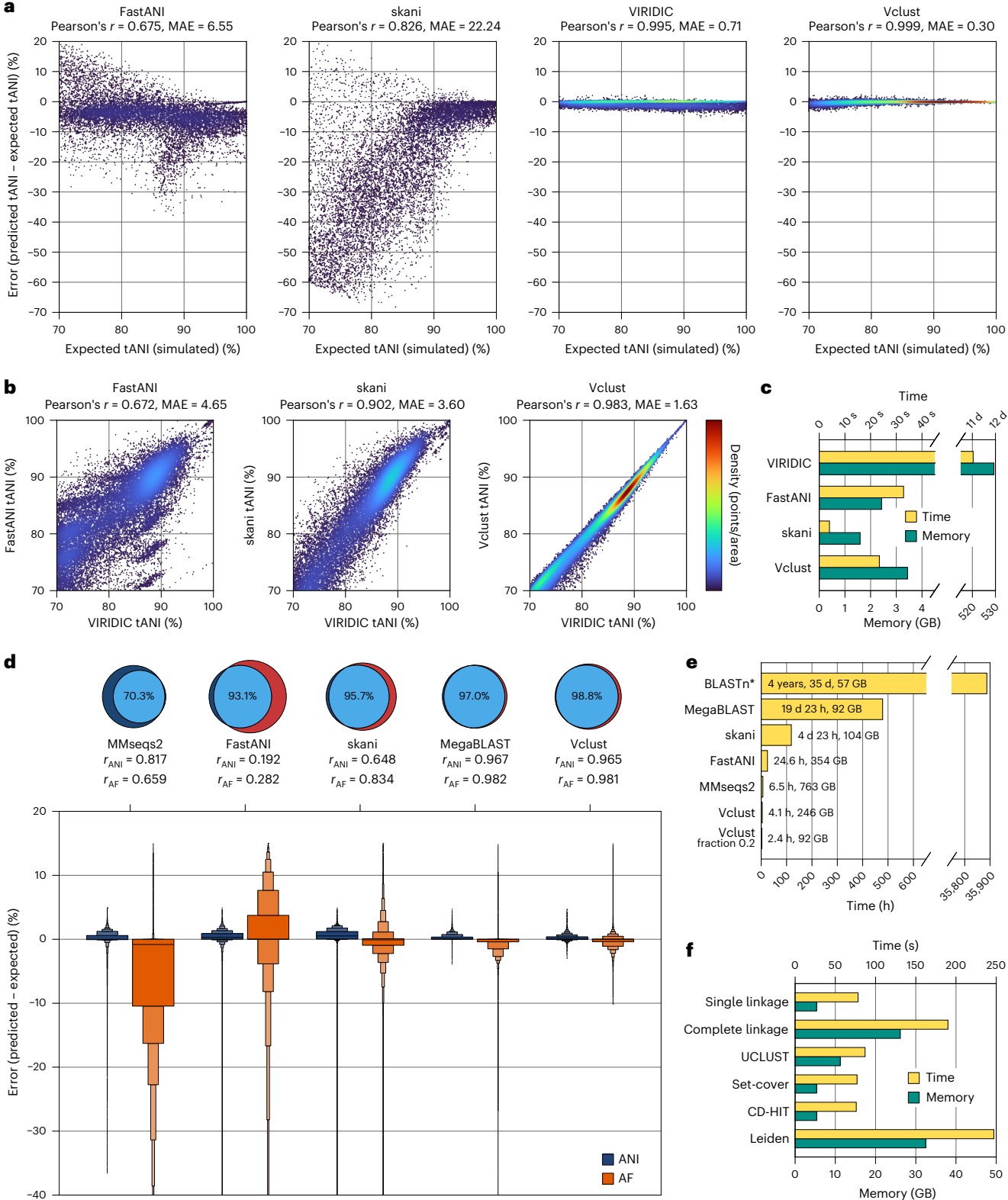

**Fig. 2 | Comparison of Vclust with other tools on various datasets. a**, Difference between predicted and expected tANI values for 10,000 bacteriophage genome pairs with simulated mutation events. **b**, Correlations with VIRIDIC tANI values for 22,607 complete bacteriophage genome pairs. **c**, Wall time and peak memory usage for processing 4,244 bacteriophage genomes (32 threads). Vclust and VIRIDIC include clustering, while FastANI and skani only calculate ANI. **d**, Venn diagrams comparing numbers of contig pairs meeting MIUViG thresholds (ANI ≥ 95% and AF ≥ 85%) predicted by BLASTn (purple) and other tools (red). The boxen plot shows the error distribution of predicted ANI and AF values relative to corresponding BLASTn-based reference values for 4,361,743 contig pairs meeting MiUVUG thresholds. The center line denotes the median, while each box level from the median contains half of the remaining observations. **e**, Wall time and peak memory usage for calculating ANI and AF among 15,677,623 IMG/VR contigs (64 threads). BLASTn values were estimated from a random sample of 1,000 query contigs. Vclust was tested in its default setting and with a 0.2 fraction of $k$-mers used at the 'prefilter' step. **f**, Wall time and peak memory usage of Vclust's clustering algorithms for grouping IMG/VR contigs into vOTUs.

respectively, by analyzing 20% of the *k*-mers in each genome during prefiltering (Fig. 2e), with negligible impact on sensitivity and specificity (Extended Data Fig. 4).

In conclusion, Vclust surpasses the current state-of-the-art methods in viral genome comparison in both accuracy and speed, remaining effective in datasets of millions of sequences. It provides a complete solution for calculating intergenomic similarities and clustering complete, partial and circularly permuted (Extended Data Fig. 5) virus genomes using various ANI measures and clustering algorithms. Given the astonishing diversity of viruses in metagenomic data, we believe that Vclust will be essential for large-scale dereplication and taxonomic classification of viral sequences. It is freely available on GitHub, with a web service option for smaller projects (https://www.vclust.org/), and its core components—Kmer-db, LZ-ANI and Clusty—are available as stand-alone tools for broader applications in sequence comparison and general clustering tasks. Similar to other tools[6], Vclust's performance may decrease with large datasets of highly similar genomes owing to the high number of sequence pairs requiring alignment and clustering after prefiltering (Methods). Future work will focus on improving scalability for large homogeneous datasets, including bacterial genomes, and implementing amino acid-based computations (for example, average amino acid identity).

## Online content

## References

1. Roux, S. et al. Minimum Information about an Uncultivated Virus Genome (MIUViG). *Nat. Biotechnol.* **37**, 29–37 (2019).
2. Camargo, A. P. et al. IMG/VR v4: an expanded database of uncultivated virus genomes within a framework of extensive functional, taxonomic, and ecological metadata. *Nucleic Acids Res.* **51**, D733–D743 (2023).
3. Nayfach, S. et al. CheckV assesses the quality and completeness of metagenome-assembled viral genomes. *Nat. Biotechnol.* **39**, 578–585 (2021).
4. Moraru, C., Varsani, A. & Kropinski, A. M. VIRIDIC-a novel tool to calculate the intergenomic similarities of prokaryote-infecting viruses. *Viruses* **12**, 1268 (2020).
5. Jain, C., Rodriguez-R, L. M., Phillippy, A. M., Konstantinidis, K. T. & Aluru, S. High throughput ANI analysis of 90K prokaryotic genomes reveals clear species boundaries. *Nat. Commun.* **9**, 5114 (2018).
6. Shaw, J. & Yu, Y. W. Fast and robust metagenomic sequence comparison through sparse chaining with skani. *Nat. Methods* **20**, 1661–1665 (2023).
7. Deorowicz, S., Gudys, A., Dlugosz, M., Kokot, M. & Danek, A. Kmer-db: instant evolutionary distance estimation. *Bioinformatics* **35**, 133–136 (2019).
8. Altschul, S. F. et al. Gapped BLAST and PSI-BLAST: a new generation of protein database search programs. *Nucleic Acids Res.* **25**, 3389–3402 (1997).
9. Steinegger, M. & Söding, J. MMseqs2 enables sensitive protein sequence searching for the analysis of massive data sets. *Nat. Biotechnol.* **35**, 1026–1028 (2017).

## Methods

### Overview

Vclust is a workflow that introduces and integrates three tools:

1. Kmer-db 2: performs the initial $k$-mer-based estimation of sequence identity of all genome pairs ('Sequence identity estimation: Kmer-db 2').
2. LZ-ANI: aligns sequence pairs with nucleotide identity exceeding a specified threshold and calculates ANI and AF measures ('Sequence alignment: LZ-ANI' and 'Calculating ANI and AF').
3. Clusty: clusters sequences based on ANI and/or AF criteria ('Clustering sequences: Clusty').

We implemented Kmer-db 2, LZ-ANI and Clusty in C++20 as stand-alone tools, adaptable for various sequence comparison and clustering tasks ('Code availability'). Vclust, a Python script, integrates these tools to calculate and cluster viral genomic sequences ('Vclust implementation').

### Sequence identity estimation: Kmer-db 2

Kmer-db 2 is an updated tool for $k$-mer-based estimation of pairwise similarities among nucleotide sequences, using either all or a selected fraction of $k$-mers. Unlike fixed-sized sketching (used, for example, by Mash[10]), Kmer-db 2 retains a proportional fraction of $k$-mers per genome, preserving the relationship between sequence lengths.

Kmer-db 2 introduces several improvements enabling the processing of tens of millions of sequences. First, unlike its predecessor, which stored similarity values in RAM as a dense matrix[7], Kmer-db 2 uses sparse matrices that retain only nonzero elements in all-to-all pairwise genome comparison mode ('all2all-sp'), allowing it to handle large and diverse genome sets. Second, Kmer-db 2 supports genome datasets partitioned into multiple input files, each generating a separate Kmer-db database. A new mode, 'all2all-parts', calculates shared $k$-mers within and across databases, optimizing memory by loading one or two databases into RAM sequentially, although at the expense of additional computational time from repeated database loading. Third, Kmer-db 2 further minimizes RAM usage by storing only genome pairs that meet a minimum threshold of shared $k$-mers and sequence identity. Finally, all modes in Kmer-db 2 support multithreading, except for the distance calculation step, which is sufficiently fast without parallelization. Supplementary Fig. 2 shows the computational performance improvements of Kmer-db 2 over Kmer-db 1, with runtime reductions of 3× to 100× across modes and substantially lower RAM requirements.

### Sequence alignment: LZ-ANI

The LZ-ANI algorithm uses Lempel–Ziv parsing[11] to align two sequences (the query and the reference).

First, the algorithm constructs two indices (dictionaries): for anchors and seeds. The anchor index maps all $a$-mers (substrings of length $a$) from both strands of the reference sequence to their positions, while the seed index performs the same mapping for shorter $s$-mers (Fig. 1b, step 1).

Next, the query is read from left to right using a sliding window of $a$ nucleotides, moving one nucleotide at a time. The parsed $a$-mers are used to search the anchor index for matches in the reference. Upon finding an exact match, the algorithm extends it in both directions (Fig. 1b, step 2). In each direction, a window of size $aw$ slides until it encounters more than a certain number of mismatches ($am$) at a time. Then, the extensions of terminal windows are trimmed to remove poorly aligned ends until they have at least $ar$ exactly matched nucleotides. This extended anchor initiates the first 'region', which corresponds to a local alignment, and is constructed as described below.

The algorithm then moves to the next nucleotide after the extended anchor and looks for $a$-mers (anywhere in the reference) and $s$-mers (within $r$ nucleotides from the end of the extended match in the reference) in the dictionaries. Four scenarios may arise:

1. No anchor or seed is found: shift by one position in the query and repeat the process of finding a new anchor or a seed match. However, if the distance in the query between the current position and the end of the previous match exceeds $q$ nucleotides, the seed search is discontinued.
2. Only a seed match is found: extend the seed similarly to the initial anchor match, append it to the region, and continue the search for a new anchor or seed match (Fig. 1b, step 3).
3. Only an anchor match is found: close the current region and extend the anchor match to initiate a new region (Fig. 1b, step 5).
4. Both anchor and seed matches are found: select the match less likely to occur by chance, based on their lengths, seed proximity ($r$ nucleotides) and the reference sequence length, leading to either scenario 2 or 3 (Fig. 1b, step 4).

Upon closing a region, the algorithm realigns the nucleotide stretches between all the extended matches within the region (Fig. 1b, step 6). This realignment aims to maximize the number of matching nucleotides between neighboring extended matches by allowing a single multi-symbol insertion in the reference or query sequence. As a result, the region represents a local alignment containing both matched and mismatched nucleotides, along with approximated indel fragments. To remove spurious alignments, regions shorter than $g$ nucleotides are excluded from further analysis.

The LZ-ANI tool reads input sequences and stores them in RAM in a compact format with three nucleotides per byte. The tool processes sequences in parallel, with each thread comparing a reference sequence to all other sequences. By default, the tool performs all-versus-all pairwise alignments, but it can also accept a filter specifying sequence pairs to align, such as a file generated by Kmer-db (used by Vclust by default).

### Alignment parameters

LZ-ANI parameters are adjustable and were optimized for virus genome sequences (Extended Data Table 2). The default anchor length was set to 11 nucleotides, matching the BLASTn default word size, which provides greater sensitivity than MegaBLAST's 28-nucleotide word size. The remaining LZ-ANI parameters were optimized using Bayesian optimization with Gaussian process minimization. This optimization involved 100 evaluations on a dataset of 10,000 pairs of complete genomes with simulated mutations (that is, substitutions, insertions, deletions, duplications, inversions and translocations) and known expected ANI values of ≥70% (Supplementary Table 13). The default Vclust parameters were selected based on the lowest MAE between the predicted and reference tANI values. Supplementary Fig. 3 compares the length, number and identity of alignments generated by Vclust (using default parameters), BLASTn and MegaBLAST.

### Calculating ANI and AF

Similarly to BLAST-based ANI methods[3,4], LZ-ANI alignment between query ($A$) and reference ($B$) encompasses 'regions', analogous to BLAST's high-scoring segment pairs. This alignment allows direct calculation of:

- $L(A, B)$—the total length (sum) of all regions when aligning query $A$ to reference $B$, in nucleotides
- $M(A, B)$—the total number of matching nucleotides in all regions

These values are used to compute seven sequence similarity measures as follows:

1. ANI for $A$ and $B$: $\frac{M(A,B)}{L(A,B)}$
2. ANI for $B$ and $A$: $\frac{M(B,A)}{L(B,A)}$
3. AF of query $A$ to reference $B$: $\frac{L(A,B)}{|A|}$
4. AF of query $B$ to reference $A$: $\frac{L(B,A)}{|B|}$

5. Global ANI for $A$ and $B$: $\frac{M(A,B)}{|A|}$

6. Global ANI for $B$ and $A$: $\frac{M(B,A)}{|B|}$

7. Total ANI: $\frac{M(A,B)+M(B,A)}{|A|+|B|}$.

## Clustering sequences: Clusty

Clusty is a versatile package facilitating rapid clustering across diverse data types, using six algorithms: single linkage, complete linkage, UCLUST[12], greedy set cover[9], CD-HIT[13] and Leiden[14]. Our implementations of these algorithms were optimized for sparse distance matrices. A linear memory complexity with the number of distances allows the clustering of tens of millions of objects, provided the matrix remains sufficiently sparse.

Clusty uses threshold-based clustering, assigning an object to a cluster if its distance from the cluster does not exceed a user-defined threshold. Depending on the algorithm, this distance can refer to the closest, furthest or centroid member. While UCLUST, greedy set cover and CD-HIT are inherently threshold-based algorithms, single and complete linkage algorithms construct dendrograms that can be pruned at customizable distance thresholds. Clusty's sparse data representation assumes all input values to meet the distance or similarity threshold. However, the tool allows clustering data at more stringent thresholds through additional filtering of any combinations of distance/similarity values (for example, tANI, ANI and AF) and/or other measure values (for example, minimum/maximum number of alignments, minimum/maximum number of matched nucleotides). Consequently, the matrix provided to Clusty does not need to be sparse; the tool can handle dense matrices and apply filtering at the loading stage.

Clusty interprets input data as a graph, where vertices represent objects and edges represent connections. Extended Data Fig. 1 shows details of the clustering algorithms and their time complexities.

## Vclust implementation

Vclust is a Python tool integrating Kmer-db 2, LZ-ANI and Clusty for streamlined computation of intergenomic sequence similarities and clustering of viral genomes. Vclust provides three commands: 'prefilter', 'align' and 'cluster' (Fig. 1a). 'prefilter' and 'align' accept a single FASTA file containing viral genomic sequences or a directory of FASTA files (one genome per file), with support for gzipped inputs and outputs.

The 'prefilter' command uses Kmer-db 2 to screen out dissimilar genome pairs before alignment, reducing the number of genome pairs to only those with sufficient $k$-mer-based sequence similarity (that is, minimum number of common $k$-mers and/or the minimum sequence identity. Sequence identity in Kmer-db 2 is calculated similarly to ANI in Mash (1 − Mash distance) but uses the overlap coefficient[15] instead of the Jaccard index. The overlap coefficient measures the intersection size of two $k$-mer sets (representing two genomic sequences) relative to the smaller set size, rather than the union of both sets. As a result, sequence identity values in the prefiltering step are generally higher than ANI from the alignment step. This allows users to set the minimum sequence identity in prefiltering close to the final ANI threshold without risking the exclusion of relevant genome pairs; for example, if targeting an ANI threshold of 95% or higher, the minimum sequence identity can be set to approximately 0.95 (Supplementary Fig. 4).

The 'align' command uses LZ-ANI to perform pairwise sequence alignments and compute ANI and AF measures between genome pairs identified by the pre-alignment filter. If the filter is not provided, Vclust aligns all possible genome pairs. The output includes two TSV files that are used for clustering: one containing ANI measures for genome pairs and the other listing genome identifiers sorted by decreasing sequence length. Optionally, Vclust can output detailed alignment results in a TSV format similar to BLASTn/MegaBLAST, with coordinates, strand orientation, matched/mismatched nucleotides and sequence identity for each local alignment.

The 'cluster' command uses Clusty for genome clustering, allowing users to specify a similarity measure (for example, tANI, ANI) and its threshold for clustering genomes, with optional additional filtering thresholds for other similarity measures, including AF. Output includes a TSV file listing genome identifiers and numerical cluster identifiers (including identifiers for singleton genomes). Alternatively, Vclust can output representative genomes instead of numerical cluster identifiers, which is particularly useful for dereplication tasks.

## Optimizing performance for highly redundant genome datasets

Vclust is designed for dereplication and clustering of viral sequences across a range of identity values. Computational performance may decline with datasets of highly redundant genome sequences (for example, tens of thousands of sequences from the same species; Supplementary Fig. 5). In all-versus-all pairwise genome comparisons in the 'prefilter' step, the high frequency of similar sequences expands the similarity matrix, increasing memory consumption and the number of pairs to align, which in turn raises computational demands for alignment and clustering. Vclust has three additional techniques to optimize performance and mitigate excessive resource consumption. First, it partitions a dataset into smaller, equally sized batches of genome sequences using the built-in multi-fasta-split C++ tool. This option considerably reduces memory requirements of the 'prefilter' step without altering results, although it may slightly increase runtime (Extended Data Fig. 4a). Second, Vclust can limit the number of $k$-mers analyzed from each genome sequence, reducing memory usage and runtime with minimal impact on sensitivity (Extended Data Fig. 4b). Third, similarly to MMseqs2 and BLAST-based methods, Vclust's 'prefilter' can restrict the number of sequences reported per query genome by selecting those with the highest sequence identity, reducing the overall number of genome pairs passing initial similarity assessment.

## Benchmarking

**Running time.** All runtimes were benchmarked on a workstation equipped with an AMD Epyc 9554 CPU (64 cores clocked at 3.1 GHz) and 1,152 GiB (approximately 1,237 GB) RAM. Unless otherwise specified, all tools were run using 64 threads. The exact commands are shown in Supplementary Tables 8, 9 and 12.

**Evaluating tANI accuracy.** The tANI accuracy of Vclust v1.2.8, FastANI (v1.33)[5], skani (v0.2.1)[6] and VIRIDIC (v1.1)[4] was assessed using two reference sets. In both reference datasets, VIRIDIC was run with default parameters (--word_size 7, --reward 2, --penalty 3, --gapopen 5, --gapextend 2) for highly sensitive BLASTn alignments. Similarly, skani was run in its most accurate mode optimized for small sequences (--slow, --s 0, --m 200). FastANI and Vclust were run with default parameters. The first reference dataset comprised 22,606 tANI values ranging from 70% to 100%, as determined by VIRIDIC across 4,244 complete genomes of bacteriophages affiliated with the ICTV using ICTV's Virus Metadata Resource (VMR v38.3). Since FastANI and skani do not directly report tANI, their values were calculated from ANI, AF and genome lengths: tANI = (ANI$_1$ × AF$_1$ × LEN$_1$ + ANI$_2$ × AF$_2$ × LEN$_2$)/ (LEN$_1$ + LEN$_2$). The second reference set contained expected (true) tANI values in the 70–100% range, derived from 10,000 pairs of bacteriophage genomes subjected to simulated mutations, including different levels of substitution, insertion, deletion, duplication, inversion and translocation events. Specifically, we randomly selected 100 genomes from the bacteriophage dataset and generated 100 copies of each genome. For each genome copy, we introduced mutations using Mutation-Simulator (v3.0.2)[16] by randomly selecting a

combination of mutation events and their corresponding frequencies (Supplementary Table 1). The expected (true) tANI value between each copy and reference genome was determined based on the variant call format produced by Mutation-Simulator, describing the exact locations of introduced mutations and the number of altered nucleotides.

**Evaluating ANI and AF accuracy.** The ANI and AF values predicted by Vclust, FastANI, skani, MegBLAST v2.13.0+ and MMseqs2 v2fad-714b525f1975b62c2d2b5aff28274ad57466 (ref. [9]) were compared to reference ANI and AF values determined by BLASTn (v2.13.0+)[8]. Since running BLASTn on the entire IMG/VR v4.1 database was not feasible, we subsampled 94,225 viral contigs and performed an all-to-all BLASTn search to identify 4,361,743 contig pairs meeting the MIUViG thresholds (ANI ≥ 95% and AF ≥ 85%). MegaBLAST, MMseqs and BLASTn outputs were used by the anicalc script from CheckV (v1.0.3)[3] to compute ANI and AF values. Pearson correlation and MAE between the predicted and expected ANI and AF values were calculated based on the 4,361,743 contig pairs meeting MIUViG thresholds (ANI ≥ 95% and AF ≥ 85%) determined by BLASTn. Given the high level of sequence identity of the reference contig pairs, if a tool did not return a result for a given contig pair, the ANI and AF values were set to zero for that pair.

**Evaluating clusterings.** The agreement between clustering results from different tools and the reference clustering was assessed using the adjusted Rand index (ARI). ARI assesses clustering similarity by comparing the number of correct clustering overlaps and disagreements[17] against those expected by chance. An ARI of 0 indicates random assignment, while a score of 1 indicates a perfect match. We used the scikit-learn (v1.3.2)[18] implementation of the ARI.

### Reporting summary
Further information on research design is available in the Nature Portfolio Reporting Summary linked to this article.

## Data availability
The datasets generated in this study have been deposited in in Figshare (https://doi.org/10.6084/m9.figshare.28294805)[19] and include complete RefSeq and GenBank genomes of 4,244 bacteriophages classified by ICTV, RefSeq and GenBank genome sequences of 10,000 bacteriophages with simulated mutations and corresponding expected total ANI values, and 94,225 metagenomic viral contigs sampled from IMG/VR v4.1 with expected BLASTn-based ANI and AF values. Supporting data generated in this study are provided in the Supplementary Information. Other databases used in the study include IMG/VR v.4.1 (https://genome.jgi.doe.gov/portal/IMG_VR/) and Virus Metadata Resource v38.3 from ICTV (https://ictv.global/vmr/). Source data are provided with this paper.

## Code availability
Vclust is available as a stand-alone tool at https://github.com/refresh-bio/vclust/ and https://doi.org/10.5281/zenodo.14756166 (ref. [20]) and as a web service at https://www.vclust.org/. Kmer-db 2, LZ-ANI and Clusty are available in the GitHub repositories at https://github.com/refresh-bio/kmer-db/, https://github.com/refresh-bio/LZ-ANI/ and https://github.com/refresh-bio/clusty/, respectively.

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

## Acknowledgements
This work is supported by the National Science Centre, Poland, project DEC-2022/45/B/ST6/03032 (to A.G. and S.D.), the European Research Council (ERC) Consolidator grant 865694: DiversiPHI, the Deutsche Forschungsgemeinschaft (DFG, German Research Foundation) under Germany's Excellence Strategy—EXC 2051—Project-ID 39071386 (to B.E.D.), the European Union's Horizon 2020 research and innovation program, under the Marie Skłodowska-Curie Actions Innovative Training Networks grant agreement no. 955974 (VIROINF; to B.E.D.), the Alexander von Humboldt Foundation in the context of an Alexander von Humboldt-Professorship founded by German Federal Ministry of Education and Research (to B.E.D. and P.R.) and the Polish Ministry of Science and Higher Education under the program 'Perły Nauki', project number PN/01/0063/2022 (to P.R.). The computations were partially performed at the Poznan Supercomputing and Networking Center (grant numbers pl0243-01 and pl0074-02).

## Author contributions
A.Z., A.G., J.B. and S.D. designed the study. A.Z. conducted the comparative analyses and developed the web service. A.G. designed and developed Clusty with input from K.S. and S.D. A.G. and S.D. designed and developed Kmer-db. S.D. designed and developed LZ-ANI. A.Z., A.G. and S.D. developed the Vclust tool. A.Z. and J.B. compared predictions to the ICTV taxonomy and reviewed ICTV proposals. A.Z., J.B., A.G., P.R., B.E.D. and S.D. analyzed the results. A.Z., A.G. and P.R. designed figures, with inputs from S.D., J.B., K.S. and B.E.D. A.Z., A.G. and S.D. wrote the manuscript with substantial contributions from B.E.D., J.B. and P.R. All authors reviewed and approved the manuscript.

## Funding

## Competing interests
The authors declare no competing interests.

## Additional information
**Extended data** is available for this paper at https://doi.org/10.1038/s41592-025-02701-7.

**Correspondence and requests for materials** should be addressed to Bas E. Dutilh or Sebastian Deorowicz.

**Single linkage**

$O(n + e)$

Hierarchical agglomerative clustering with a distance between groups defined as a distance between their closest members. It corresponds to finding all consistent subgraphs in a graph and is performed using breadth-first search. Equivalent to MMseqs2 mode 1 clustering.

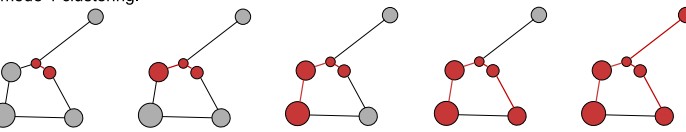

**Complete linkage**

$O(n + e \log e)$

Hierarchical agglomerative clustering with a distance between groups defined as a distance between their furthest members. Equivalent to finding a disjoint set of complete subgraphs covering the entire graph. Fast identification of clusters to merge is performed by storing distances in a heap.

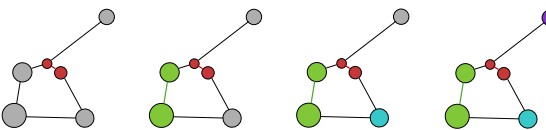

**UCLUST**

$O(n + e)$

Greedy clustering with objects investigated from the most to the least representative (for sequences usually from the longest to the shortest). The first object becomes a centroid; the following objects are either (a) assigned to their closest connected centroids or (b) become new centroids if they are not connected to any of the existing ones.

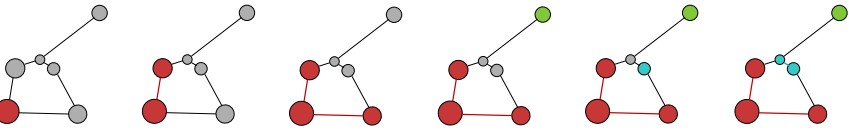

**Greedy set cover**

$O(n \log n + e)$

Greedy clustering with objects investigated in descending order with respect to the number of neighbors. Every unassigned object becomes a new centroid with all connected objects being assigned to it. Equivalent to MMseqs2 mode 0 clustering.

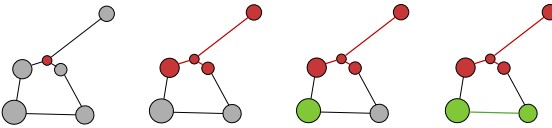

**CD-HIT**

$O(n + e)$

Greedy clustering with objects investigated from the most to the least representative (for sequences usually from the longest to the shortest). Every unassigned object becomes a new centroid with all connected objects being assigned to it. Equivalent to MMseqs2 mode 2 clustering.

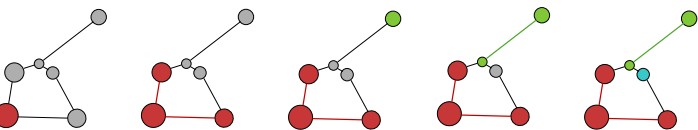

**Leiden**

Undetermined

Iterative heuristic for finding communities in networks. It consists of three phases: (1) local moving of nodes, (2) refinement of the partition, and (3) aggregation of the network using the refined and the non-refined partitions.

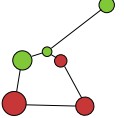

- ⬤ Not clustered yet
- 🔴 Cluster 1
- 🟢 Cluster 2
- 🔵 Cluster 3
- 🟣 Cluster 4

**Extended Data Fig. 1 | Clustering algorithms in Clusty.** The dataset is represented as a graph: objects (genomes) are shown as vertices with sizes proportional to user-assigned weights (for example, sequence length) and edges indicate distances (for example, 1 − tANI). In the consecutive steps of each algorithm objects are assigned to clusters marked with distinct colors. The left panel shows each algorithm's time complexity, with $n$ as the number of vertices and $e$ as the number of edges. Distance values for this example are available in the source data provided with the paper.

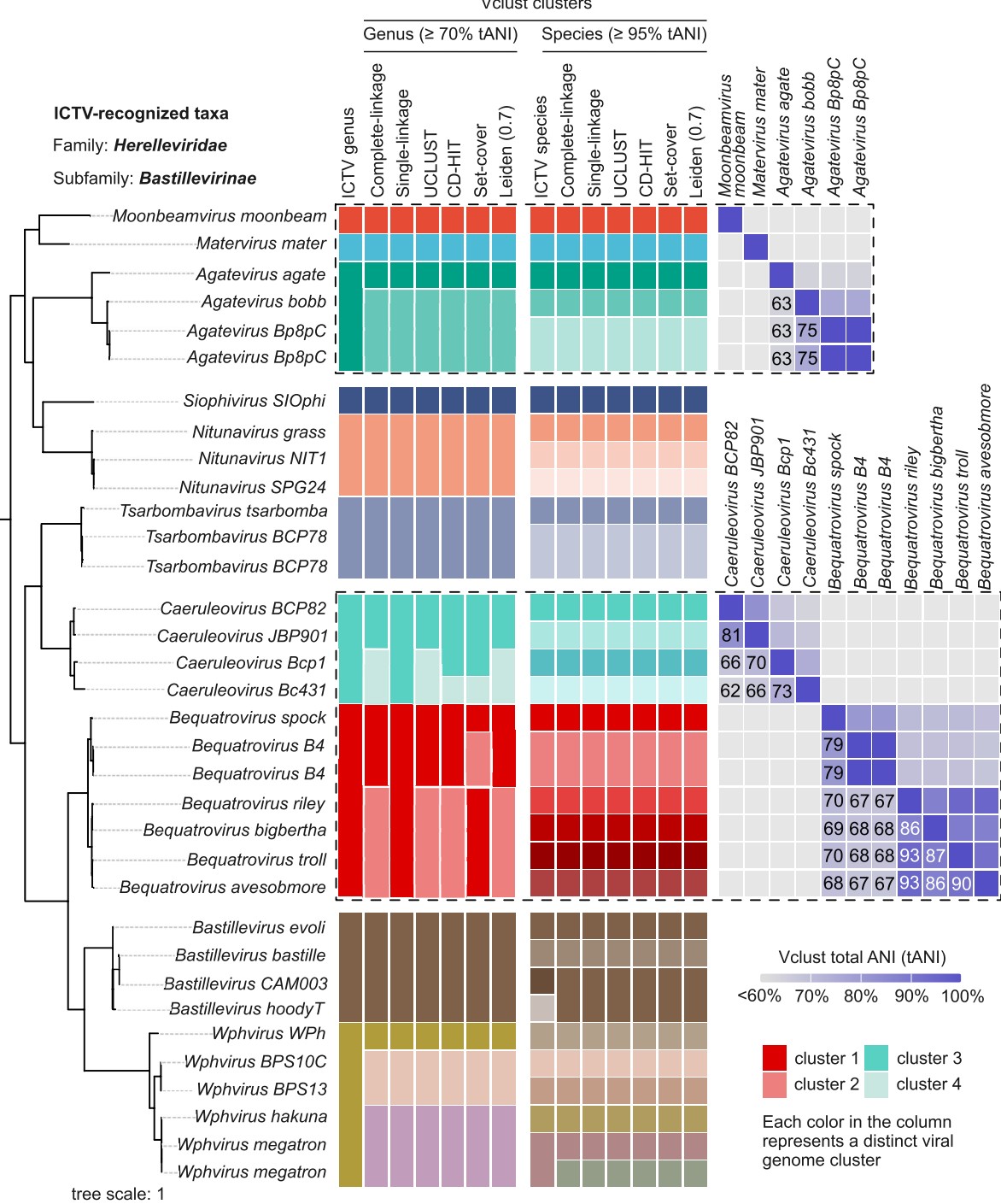

**Extended Data Fig. 2 | Vclust application to ICTV-recognized genera (threshold 70% tANI) and species (threshold 95% tANI).** The subtree of *Bastillevirinae* branch from the maximum-likelihood tree based on a concatenated alignment of 10 marker proteins in *Herelleviridae* family. Leaves are labeled with species names and four species are represented by more than one genome (for example, multiple strains). Snippets of the total heatmap on the right highlight instances where Vclust clustering algorithms and/or ICTV disagree. Numbers in these boxes show tANI (only values higher than 60% - Vclust reliability threshold are shown). Despite recommended rank-demarcation criteria, the classification of viruses is based on multiple forms of evidence, ranging from calculating tANI between pairs to protein content and phylogenetic analyses. Therefore, inconsistencies between Vclust clustering and the official ICTV taxonomy may arise, especially if the tANI oscillates around the demarcation criterion, such as between *Bastillevirus hoodyT* and *Bastillevirus evoli*, where the tANI is 95.2%.

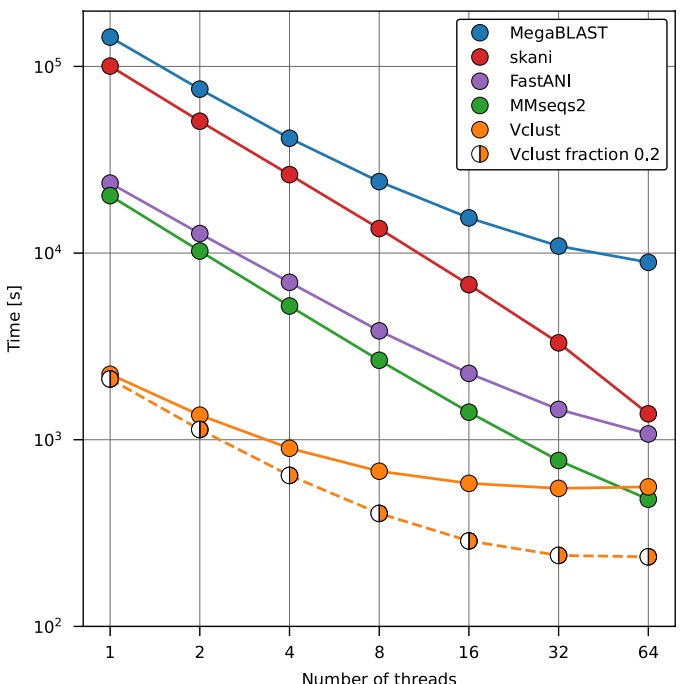
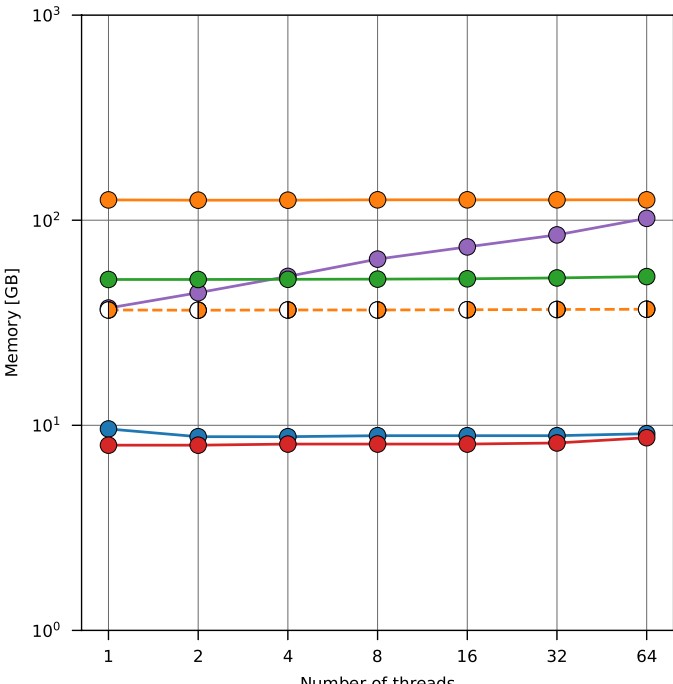

**Extended Data Fig. 3 | Wall time and peak memory usage of Vclust and other tools as a function of an increasing number of CPU threads on a dataset of 1 million metagenomic viral contigs.** The dataset was sampled from IMG/VR v4.1, and the execution commands for the tools are detailed in Supplementary Table 12. Vclust was tested with all $k$-mers (default) and 0.2 fraction of $k$-mers retained at the *prefilter* step. The left panel shows execution times. The runtimes of all tools decrease linearly with up to eight threads, but beyond this, FastANI, MegaBLAST, and Vclust showed reduced speed-up while skani and MMseqs2

maintained linear scaling. Vclust with 0.2 $k$-mers fraction was the fastest algorithm independently of the number of threads. It was followed by the default Vclust setting with the exception of 64 threads, where slow-down caused by the task fragmentation overhead at the *prefilter* step made it inferior by a small margin to less accurate MMseqs2. In the right panel, memory usage was constant across thread counts for all algorithms except FastANI, which showed a linear increase in memory consumption.

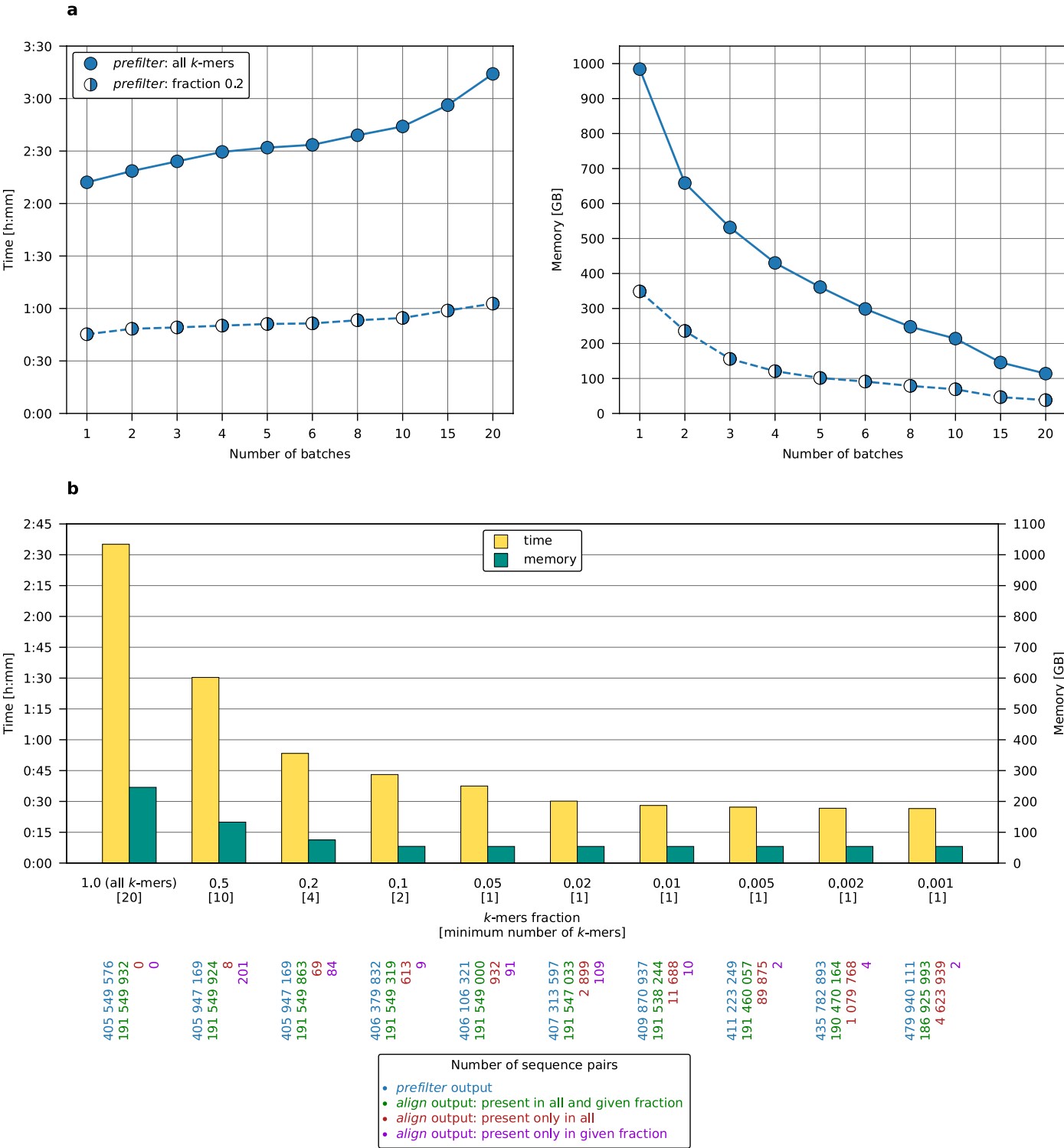

**Extended Data Fig. 4 | Performance of Vclust's prefiltering step using the IMG/VR v4.1 dataset of 15,677,623 viral contigs. a**, Wall time and peak memory usage for prefiltering genomes in a varying number of equally-sized sequence batches. The prefilter was configured to consider all *k*-mers (default) or 0.2 fraction of *k*-mers. Increasing the number of batches slightly increased runtime, significantly reducing memory usage for both *k*-mer fractions. **b**, Wall time and peak memory usage for different combinations of *k*-mers fractions (1 to 0.001) and minimum shared *k*-mers (20 to 1) during prefiltering. Below each combination, the outputs of *prefilter* and *align* steps are compared to the default settings (*k*-mers fraction 1 and minimum shared *k*-mers 20). The numbers represent: contig pairs passing the *prefilter* step (≥95% sequence identity; blue), pairs passing the *align* step (ANI ≥ 95%, AF ≥ 85%) common to the pairs obtained using all *k*-mers (green), contig pairs missed compared to using all *k*-mers (red), and additional contig pairs detected at the specified *k*-mers fraction but missing compared to using all *k*-mers (magenta). For instance, a 0.2 *k*-mers fraction reduces time and memory usage threefold, with a minimal impact on sensitivity and specificity.

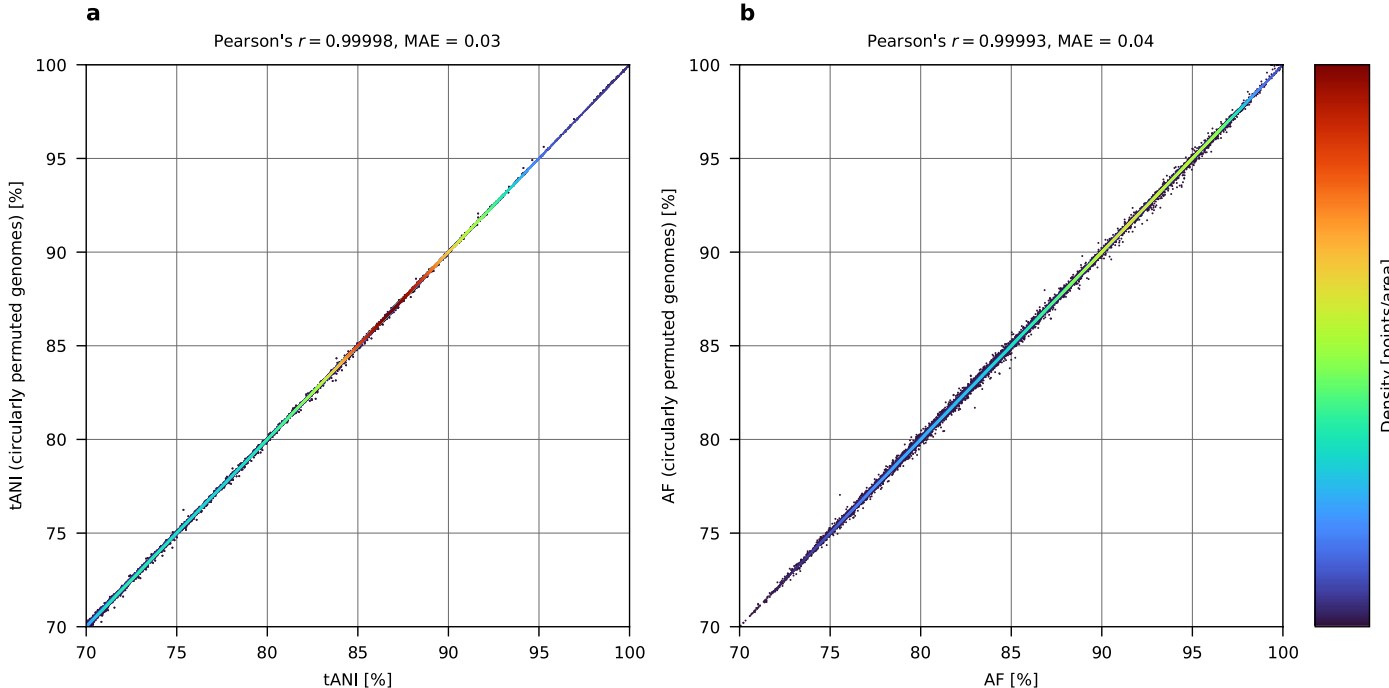

**Extended Data Fig. 5 | Evaluation of Vclust's accuracy on circularly permuted genomes.** Comparison of **a**, total average nucleotide identity (tANI) and **b**, aligned fraction (AF) values on a reference dataset of 4,244 complete bacteriophage genomes and the same dataset where two regions in each genome were swapped at a random breakpoint position. Both plots cover 40,618 genome pairs with tANI ≥ 70%.

**Extended Data Table 1 | Comparison of Vclust features with other tools for sequence comparison and clustering of complete or metagenome-assembled virus genomes**

| Feature \ Tool | Vclust | VIRIDIC | anicalc (CheckV) | MMseqs2 | FastANI | skani |
|---|---|---|---|---|---|---|
| Average Nucleotide Identity (ANI) estimation method | *k*-mer-based prefilter (Kmer-db) followed by alignment (LZ-ANI) | alignment (BLASTn) | alignment (BLASTn / MegaBLAST / MMseqs2) | *k*-mer matching as a prefilter followed by alignment | *k*-mer-based identification of local alignments (Mashmap) | approximate alignment with sparse *k*-mer chaining |
| Report: | | | | | | |
| - ANI[a] | yes | - | yes | - | yes | yes |
| - Global ANI[b] | yes | - | - | - | - | - |
| - Total ANI[c] | yes | yes | - | - | yes* | yes* |
| - Aligned fraction (AF)[d] | yes | - | yes | - | yes | yes |
| - Number of segments aligned | yes | - | yes | - | - | - |
| Recommended ANI to report | ≥ 70% | ≥ 70% | N/A | N/A | ≥ 80% | ≥ 82% |
| Clustering algorithms | yes | yes | yes | yes | - | - |
| - Single-linkage | yes | yes | - | yes | - | - |
| - Complete-linkage | yes | yes | - | - | - | - |
| - Leiden algorithm | yes | - | yes | - | - | - |
| - CD-HIT | yes | - | - | yes | - | - |
| - UCLUST | yes | - | - | - | - | - |
| - Greedy set cover | yes | - | - | yes | - | - |
| Clustering thresholds: | | | | | | |
| - ICTV[e] | yes | yes | - | - | - | - |
| - MIUViG[f] | yes | - | yes | yes | yes | yes |
| Web service | yes | yes | - | - | - | - |
| Summary | Very accurate, large scale | Very accurate, small scale (≤ 300 complete genomes) | Accurate, medium and large scale | Large scale, no ANI reporting | Less accurate than alignment, large scale, no clustering | Less accurate than alignment, large scale, no clustering |

[a]Number of identical bases across local alignments between two genomes divided by the total length of these alignments [b]Number of identical bases across local alignments between two genomes divided by the length of the query/target genome [c]Number of identical bases in local alignments between query-target and target-query genomes by the sum length of both genomes. This is equivalent to intergenomic similarity in VIRIDIC. [d]Fraction of query/target genome aligned to one another [e]International Committee on Taxonomy of Viruses (ICTV) recommends assigning viruses to the same species or genus based on total ANI of 95% and 70%, respectively. [f]Minimum Information about an Uncultivated Virus Genome (MIUViG) standards recommend assigning viruses to virus operational taxonomic units based on 95% ANI over 85% alignment fraction (relative to the shorter sequence). *Although total ANI is not directly reported, it can be approximated from ANI, alignment fraction, and genome lengths.

**Extended Data Table 2 | Parameters of LZ-ANI sequence aligner. All length values are given in nucleotides**

| Parameter | Role | Default value |
|---|---|---|
| *a* | anchor length | 11 |
| *s* | seed length | 7 |
| *r* | maximum distance between approx. matches in reference | 40 |
| *q* | maximum distance between approx. matches in query | 40 |
| *g* | minimum region length to consider | 35 |
| *aw* | window length for match (anchor or seed) extension | 15 |
| *am* | maximum number of mismatches in the extension window | 7 |
| *ar* | minimum number of consecutive matches at the end of the extension | 3 |

|---|---|
| | Bas E. Dutilh |

# Reporting Summary

## Statistics

For all statistical analyses, confirm that the following items are present in the figure legend, table legend, main text, or Methods section.

| n/a | Confirmed | |
|---|---|---|
| ☐ | ☒ | The exact sample size (*n*) for each experimental group/condition, given as a discrete number and unit of measurement |
| ☐ | ☒ | A statement on whether measurements were taken from distinct samples or whether the same sample was measured repeatedly |
| ☒ | ☐ | The statistical test(s) used AND whether they are one- or two-sided<br>*Only common tests should be described solely by name; describe more complex techniques in the Methods section.* |
| ☒ | ☐ | A description of all covariates tested |
| ☒ | ☐ | A description of any assumptions or corrections, such as tests of normality and adjustment for multiple comparisons |
| ☐ | ☒ | A full description of the statistical parameters including central tendency (e.g. means) or other basic estimates (e.g. regression coefficient) AND variation (e.g. standard deviation) or associated estimates of uncertainty (e.g. confidence intervals) |
| ☒ | ☐ | For null hypothesis testing, the test statistic (e.g. *F*, *t*, *r*) with confidence intervals, effect sizes, degrees of freedom and *P* value noted<br>*Give P values as exact values whenever suitable.* |
| ☒ | ☐ | For Bayesian analysis, information on the choice of priors and Markov chain Monte Carlo settings |
| ☒ | ☐ | For hierarchical and complex designs, identification of the appropriate level for tests and full reporting of outcomes |
| ☐ | ☒ | Estimates of effect sizes (e.g. Cohen's *d*, Pearson's *r*), indicating how they were calculated |

*Our web collection on statistics for biologists contains articles on many of the points above.*

## Software and code

Policy information about availability of computer code

| Data collection | No software was used. |
|---|---|
| Data analysis | Our tool Vclust 1.2.8 (described in the manuscript and available at https://github.com/refresh-bio/vclust), integrating Kmer-db 2.2.3 (https://github.com/refresh-bio/kmer-db), LZ-ANI 1.2.2 (https://github.com/refresh-bio/lz-ani), and Clusty 1.1.3 https://github.com/refresh-bio/clusty, was used for data analysis. Benchmarking was performed against Viridic 1.1, FastANI 1.33, skani 0.2.1, MMseqs2 v2fad7, MegaBLAST 2.13+, and BLASTn 2.13+, and the anicalc Python script from CheckV 1.0.3. Simulated datasets of bacteriophage genomes were prepared using Mutation-Simulator 3.0.2. The adjusted Rand Index was calculated with scikit-learn v1.3.2. |

For manuscripts utilizing custom algorithms or software that are central to the research but not yet described in published literature, software must be made available to editors and reviewers. We strongly encourage code deposition in a community repository (e.g. GitHub). See the Nature Portfolio guidelines for submitting code & software for further information.

## Data

The datasets generated in this study have been deposited in in Figshare (https://doi.org/10.6084/m9.figshare.28294805) and include complete RefSeq and GenBank genomes of 4,244 bacteriophages classified by ICTV, RefSeq and GenBank genome sequences of 10,000 bacteriophages with simulated mutations and corresponding expected total ANI values, and 94,225 metagenomic viral contigs sampled from IMG/VR v4.1 with expected BLASTn-based ANI and AF values. Supporting data generated in this study are provided in the Supplementary Information, Source Data and Supplementary Data files. Other databases used in the study include IMG/VR v.4.1 (https://genome.jgi.doe.gov/portal/IMG_VR/) and Virus Metadata Resource v38.3 from ICTV (https://ictv.global/vmr). Source data are provided with this paper.

## Human research participants

Policy information about studies involving human research participants and Sex and Gender in Research.

| Reporting on sex and gender | N/A (not human research) |
|---|---|
| Population characteristics | N/A (not human research) |
| Recruitment | N/A (not human research) |
| Ethics oversight | N/A (not human research) |

Note that full information on the approval of the study protocol must also be provided in the manuscript.

# Field-specific reporting

Please select the one below that is the best fit for your research. If you are not sure, read the appropriate sections before making your selection.

☒ Life sciences　　☐ Behavioural & social sciences　　☐ Ecological, evolutionary & environmental sciences

For a reference copy of the document with all sections, see nature.com/documents/nr-reporting-summary-flat.pdf

# Life sciences study design

All studies must disclose on these points even when the disclosure is negative.

| Sample size | Sample-size calculations were not performed. In most experiments, the entire datasets were used, with the following exceptions:<br>* For estimating BLASTN runtime on the IMG/VR dataset, 1,000 contigs were randomly selected.<br>* For the simulated dataset, 100 bacteriophage genomes were randomly chosen from the ICTV dataset, and 100 variants were generated for each using Mutation-Simulator.<br>* For estimating ANI/AF accuracy, 94,225 viral contigs were randomly selected from the IMG/VR dataset. |
|---|---|
| Data exclusions | No data were excluded from the analyses. Complete datasets served as benchmarks. Additionally, for one reference dataset containing bacteriophage genomes from ICTV, we manually created a subset of high-quality species groupings by excluding genome pairs without sufficient evidence in ICTV's proposals supporting their classification as a single species. The excluded genome pairs are listed in Tables S4 and S5. |
| Replication | Testing was conducted on various machine configurations, including Linux and Windows x64-based desktops, Linux x64-based servers, MacOS x64-based desktops, and MacOS ARM-based desktops. The results were consistent across all configurations. |
| Randomization | We propose a new method for calculating ANI and clustering. All methods were applied to all datasets, except for BLASTN on the entire IMGVR dataset, which was infeasible due to an estimated runtime of over four years. Therefore, for estimating BLASTN runtime, we randomly selected 1,000 contigs and extrapolated the results. |
| Blinding | Not applicable, as this paper focuses on method development and benchmarking. |

# Reporting for specific materials, systems and methods

We require information from authors about some types of materials, experimental systems and methods used in many studies. Here, indicate whether each material, system or method listed is relevant to your study. If you are not sure if a list item applies to your research, read the appropriate section before selecting a response.

## Materials & experimental systems

| n/a | Involved in the study |
|-----|----------------------|
| ☒ ☐ | Antibodies |
| ☒ ☐ | Eukaryotic cell lines |
| ☒ ☐ | Palaeontology and archaeology |
| ☒ ☐ | Animals and other organisms |
| ☒ ☐ | Clinical data |
| ☒ ☐ | Dual use research of concern |

## Methods

| n/a | Involved in the study |
|-----|----------------------|
| ☒ ☐ | ChIP-seq |
| ☒ ☐ | Flow cytometry |
| ☒ ☐ | MRI-based neuroimaging |

