## [Peer Review File · Nature Methods]

Ultrafast and accurate sequence alignment and clustering of viral genomes

Corresponding Author: Professor Bas Dutilh

Version 0:

Decision Letter:

8th Aug 2024

Dear Professor Deorowicz,

Your Brief Communication, "Ultrafast and accurate sequence alignment and clustering of viral genomes", has now been seen by 3 reviewers (Reviewers 2 and 3 co-review it). As you will see from their comments below, although the reviewers find your work of potential interest, they have raised a number of concerns. We are interested in the possibility of publishing your paper in Nature Methods, but would like to consider your response to these concerns before we reach a final decision on publication.

We therefore invite you to revise your manuscript to fully address all these concerns.

Link Redacted

We hope to receive your revised paper within 4 months. If you cannot send it within this time, please let us know. In this event, we will still be happy to reconsider your paper at a later date so long as nothing similar has been accepted for publication at Nature Methods or published elsewhere.

OPEN SCIENCE REQUIREMENTS

REPORTING SUMMARY AND EDITORIAL POLICY CHECKLISTS

DATA AVAILABILITY

All novel DNA and RNA sequencing data, protein sequences, genetic polymorphisms, linked genotype and phenotype data, gene expression data, macromolecular structures, and proteomics data must be deposited in a publicly accessible database, and accession codes and associated hyperlinks must be provided in the "Data Availability" section.

CODE AVAILABILITY

Please include a "Code Availability" subsection in the Online Methods which details how your custom code is made available. Only in rare cases (where code is not central to the main conclusions of the paper) is the statement "available upon request" allowed (and reasons should be specified).

MATERIALS AVAILABILITY

ORCID

Nature Methods is committed to improving transparency in authorship. As part of our efforts in this direction, we are now requesting that all authors identified as 'corresponding author' on published papers create and link their Open Researcher and Contributor Identifier (ORCID) with their account on the Manuscript Tracking System (MTS), prior to acceptance. This applies to primary research papers only. ORCID helps the scientific community achieve unambiguous attribution of all scholarly

contributions. You can create and link your ORCID from the home page of the MTS by clicking on 'Modify my Springer Nature account'. For more information please visit www.springernature.com/orcid.

Sincerely,

Lin Tang, PhD
Senior Editor
Nature Methods

Reviewers' Comments:

Reviewer #1 (Remarks to the Author):

In this brief note, Zielezinski et al describe Velust, an ultrafast computational method for clustering viral genome sequences by sequence similarity. The description seems convincing, given the format, and considering the combination of accuracy and speed, I do believe the method will be highly useful.

I have no major criticisms, only a few questions that can be answered quite easily:

-I do not see why this method is specifically applicable to viral sequences. It seems quite general unless I am missing some idiosyncratic features.

-In 1160-62, I am rather confused by the mention of sequences with simulated mutations. What about real genomes?

-In the Online Methods (11 243-245), it is emphasized that Velust compares alignable regions. I think, however, that this is a key feature that is best pointed out in the Main.

-176 says that "all tools showed low agreement with ICTV". However, I think in the cases of Velust and VIRIDIC, the agreement is actually quite good, given the inconsistencies that are inevitable in ICTV assignments

Reviewer #2 (Remarks to the Author):

Vclust offers a novel approach to viral genome analysis by using Lempel-Ziv parsing to accurately calculate average nucleotide identity (ANI). It efficiently clusters millions of viral genomes, outperforming existing methods in both accuracy and speed. Vclust integrates an enhanced k-mer database and multiple clustering algorithms, to handle large-scale metagenomic datasets effectively. Overall the program is easy to use and the manuscript is well-written. However we raise several issues below that we hope the authors can address.

[major comments]

The authors include a number of clustering algorithms implemented by Clusty. However, there is no detailed comparison of these clustering algorithms. In particular, how accurate are the different methods for species delineation? What are the optimal cutoffs (ANI/AF) and clustering algorithm for identifying vOTUs (MIUVIG definition and ICTV definition). More generally, how do intra-cluster and inter-cluster distances vary across each clustering method? Also, Figure 1c is not particularly helpful for explaining how the various clustering algorithms work.

It has also been suggested that ICTV-genera for phages should be determined using an ANI threshold of >70% (10.3390/v13030506). Given that Vclust appears to perform well at lower ANI values, it would be useful to include some analyses examining its ability to cluster phage genomes at this threshold. How accurate are the different methods for genus delineation, and what are the optimal parameters?

Accuracy of Vclust should be evaluated across varying genome size, ranging from the smallest (e.g. <1 Kb) to largest (e.g. >1 Mb) viruses. Also, some tools fail to accurately calculate AF when the target genome is circularly permuted (e.g. CD-HIT) – do Vclust and other tools suffer from this same issue?

The authors should perform a more detailed scalability analysis. How do RAM and CPU scale with increasing # of genomes? How does RAM and CPU scale with relatedness of genomes (e.g. very dense vs very sparse graphs)?

How does the prefiltering step affect Vclust's speed/scalability? Is prefiltering to a –min-ident of 0.7 necessary for vOTU clustering (identifying all genome pairs with >95% ANI)? Or can this value be raised to decrease the pairwise comparisons performed by the align step? How accurate is the prefilter step relative to the align step?

Can the authors please provide data supporting their choice in default parameters? Anchor length, seed length, window length, etc. should be included.

The performance of Vclust should be benchmarked vs other tools using different #s of CPUs (e.g. 1, 2, 4, 8, 16, 32). In Figure 2c Vclust is compared to other tools using 32 CPUs, indicating that MEGABLAST would take >4 years of CPU time to process IMG/VR v4. It is known that tools like MegaBLAST scale poorly with the number of CPUs.

[minor comments]

The same set of tools should be evaluated in all benchmarks. CheckV's ANIcalc is left out of initial benchmarks but included in latter ones. Please report performance in the text and in figure a (panels a, b, & c)

The authors claim that Kmer-db2 outperforms Kmer-db1, but no data is given to support this claim.

[Line 101-102] Language / typo

[Figure 2a] What parameters were used for VIRIDIC? The VIRIDIC commands for 2b and 2c are given in the supplementary tables, but this is not given for Fig 2a.

[Table S8] "Notably, vOTU clusters obtained from Vclust and MegaBLAST exhibited high adjusted rand scores". This logic is circular since IMG/VR vOTUs were generated using MegaBLAST + Leiden clustering.

[Github] Tool documentation and setup is fairly straightforward, but creating a BioConda package would greatly increase this tool's accessibility and usage.

[vClust CLI] An option to output the alignment coordinates in addition to the number of alignments could allow more detailed comparison of the differences between vClust and alternative alignment tools. An exploration of alignments identified by vClust that were missed by alternative alignment tools would be very informative (how many of these regions are there? what is the ANI of these regions? Why might these regions be missed by megaBLAST/blastn?)

Editor: Reviewer #3 co-reviews with Reviewer #2

Version 1:

Decision Letter:

Our ref: NMETH-BC57097A

1st Jan 2025

Dear Dr. Dutilh,

Thank you for submitting your revised manuscript "Ultrafast and accurate sequence alignment and clustering of viral genomes" (NMETH-BC57097A). It has now been seen by the original referees and their comments are below. The reviewers find that the paper has improved in revision, and therefore we'll be happy in principle to publish it in Nature Methods, pending minor revisions to comply with our editorial and formatting guidelines.

TRANSPARENT PEER REVIEW

Please note: we allow redactions to authors' rebuttal and reviewer comments in the interest of confidentiality. If you are concerned about the release of confidential data, please let us know specifically what information you would like to have removed. Please note that we cannot incorporate redactions for any other reasons. Reviewer names will be published in the peer review files if the reviewer signed the comments to authors, or if reviewers explicitly agree to release their name. For more information, please refer to our [FAQ](https://www.nature.com/documents/nr-transparent-peer-review.pdf) page.

ORCID

Sincerely,

Lin Tang, PhD
Senior Editor
Nature Methods

Reviewer #1 (Remarks to the Author):

I find the responses to reviewers quite convincing and have no further critical comments. It will be a very useful tool.

Reviewer #2 (Remarks to the Author):

We thank the authors for carefully addressing all of the points we raised. Their effort and thorough consideration of our feedback are appreciated. We have no other major requests.

Responses to specific revisions:

Major comment 1: The data in Table S5 and Figure S1 are very insightful and address the original comments regarding clustering algorithms. The caution regarding the recommendation of a specific clustering algorithm seems appropriate in this scenario.

Major comment 2: The updated Figure 1c and Extended Data Fig. 1 do a much better job of explaining the clustering algorithms and will be very useful to vClust users when they are trying to decide on a clustering algorithm to use.

Major comment 3: The author's notes regarding inaccuracies in ICTV at the genus level are important and the data provided displays this issue well.

Major comment 4: This experiment is important and highlights that vClust's performance is maintained at low genome lengths, where other tools perform poorly. Since vClust is also likely to be used for less-similar genomes (70-95% ANI), it would also be insightful to see a similar experiment investigating how well vClust performs for genome pairs in this category.

Major comment 5: This experiment highlights vClust's ability to accurately compare circularly permuted genomes.

Major comment 6: The data presented in these two figures is particularly important due to the increasing size of viral databases that need to be aligned/clustered. The data and optimization displayed in Extended data 4 will be very useful for vClust users who are using this tool at scale.

Major comment 7: This data is very useful in setting an appropriate `--min-ident` threshold and will provide users with much better guidance and confidence in setting these thresholds. The updated documentation and examples are very helpful resources.

Major comment 8: The justification for vClust's default parameters is very strong and the inclusion of this data will aid in the transparency and confidence in this tool.

Major comment 9: This data is insightful and provides guidance for users hoping to maximize this tool's efficiency.

Minor comment 1: The author's comments and selection of a subset of tools seems appropriate.

Minor comment 2: This data highlights the substantial benefits that Kmer-db2 provides relative to Kmer-db1, further showcasing the novelty and performance of this tool.

Minor comment 3: The original comment has been addressed.

Minor comment 4: The original comment has been addressed.

Minor comment 5: The original comment has been addressed.

Minor comment 6: This is a great addition that will increase the visibility, accessibility and usage of this tool.

Minor comment 7: The original comment has been addressed.

Minor comment 8: This data is insightful and addresses the major question originally posed.

Version 2:

Decision Letter:

14th Apr 2025

Dear Professor Dutilh,

I am pleased to inform you that your Brief Communication, "Ultrafast and accurate sequence alignment and clustering of viral genomes", has now been accepted for publication in Nature Methods. The received and accepted dates will be 9th Jul 2024 and 14th Apr 2025. This note is intended to let you know what to expect from us over the next month or so, and to let you know where to address any further questions.

Over the next few weeks, your paper will be copyedited to ensure that it conforms to Nature Methods style. Once your paper is typeset, you will receive an email with a link to choose the appropriate publishing options for your paper and our Author Services team will be in touch regarding any additional information that may be required.

Once proofs are generated, they will be sent to you electronically and you will be asked to send a corrected version within 48 hours. It is extremely important that you let us know now whether you will be difficult to contact over the next month. If this is the case, we ask that you send us the contact information (email, phone and fax) of someone who will be able to check the proofs and deal with any last-minute problems.

If, when you receive your proof, you cannot meet the deadline, please inform us at rjsproduction@springernature.com immediately.

If you are active on X or Bluesky, please e-mail me your and your coauthors' handles so that we may tag you when the paper is published.

To assist our authors in disseminating their research to the broader community, our SharedIt initiative provides you with a unique shareable link that will allow anyone (with or without a subscription) to read the published article. Recipients of the link with a subscription will also be able to download and print the PDF. As soon as your article is published, you will receive an automated email with your shareable link.

Please note that you and your coauthors may order reprints and single copies of the issue containing your article through Springer Nature Limited's reprint website, which is located at <http://www.nature.com/reprints/author-reprints.html>. If there are any questions about reprints please send an email to author-reprints@nature.com and someone will assist you.

Please feel free to contact me if you have questions about any of these points. Thank you very much for publishing your paper at Nature Methods!

Best regards,

Lin Tang, PhD
Senior Editor
Nature Methods

** Visit the Springer Nature Editorial and Publishing website at [http://www.springernature.com/editorial-and-publishing-jobs?](http://www.springernature.com/editorial-and-publishing-jobs?utm_source=ejP_NMeth_email&utm_medium=ejP_NMeth_email&utm_campaign=ejp_Nmeth) www.springernature.com/editorial-and-publishing-jobs for more information about our career opportunities. If you have any questions please click [here](mailto:editorial.publishing.jobs@springernature.com).**

Response to Reviewer 1

R: In this brief note, Zielezinski et al describe Vclust, an ultrafast computational method for clustering viral genome sequences by sequence similarity. The description seems convincing, given the format, and considering the combination of accuracy and speed, I do believe the method will be highly useful. I have no major criticisms, only a few questions that can be answered quite easily.

A: We would like to thank the Reviewer for the kind words and thorough review.

R: I do not see why this method is specifically applicable to viral sequences. It seems quite general unless I am missing some idiosyncratic features.

A: The Reviewer is right - Vclust can be applied to genome sequences in general. Our focus on viral sequences stems from the lack of a single tool that effectively handles the various ANI metrics specifically recommended by different viral genomics consortia, as shown in **Extended Data Table 1**. Also, our expertise lies primarily in comparative genomics of viruses, which allowed us to validate the results more reliably, select appropriate datasets for validation, and critically review the outputs. Moreover, the Vclust parameters were optimized specifically for viral genomes and contigs. However, we recognize the growing need to handle bacterial genomes, particularly in metagenomics. Although Vclust is computationally robust and suitable for bacterial genomes, we have not yet tested its accuracy in this context. We have received numerous requests via GitHub and Twitter (X) for bacterial genome support and plan to extend Vclust's capabilities to prokaryotic genomes in the near future, as noted in the Methods section (**lines 401-403**).

R: In ll60-62, I am rather confused by the mention of sequences with simulated mutations. What about real genomes?

A: In this study, we tested Vclust using both simulated and real genomes. Virus genomes with simulated mutations provided a reference for total ANI (tANI), enabling us to compare these values with tANI predicted by VIRIDIC, Vclust, FastANI, and skani (**Fig. 2a, Tables S1-S2**). Nevertheless, most of our comparisons focus on real viral genomes (**Fig. 2b-f, Extended Data Figs. 2-8, Tables S3-S12**). For these tests, we used reference ANI values obtained through BLASTn, known for its high accuracy but orders of magnitude slower speed.

R: In the Online Methods (ll 243-245), it is emphasized that Vclust compares alignable regions. I think, however, that this is a key feature that is best pointed out in the Main.

A: Thank you for this valuable suggestion. We have added a sentence to the Main Text highlighting this feature (**lines 57-59**).

R: 176 says that "all tools showed low agreement with ICTV". However, I think in the cases of Vclust and VIRIDIC, the agreement is actually quite good, given the inconsistencies that are inevitable in ICTV assignments.

A: We agree and have revised the phrase to reflect better the level of agreement of the tools with ICTV (**lines 79-80**).

Response to Reviewers 2 and 3

R: Vclust offers a novel approach to viral genome analysis by using Lempel-Ziv parsing to accurately calculate average nucleotide identity (ANI). It efficiently clusters millions of viral genomes, outperforming existing methods in both accuracy and speed. Vclust integrates an enhanced k-mer database and multiple clustering algorithms, to handle large-scale metagenomic datasets effectively. Overall the program is easy to use and the manuscript is well-written. However we raise several issues below that we hope the authors can address.

A: We are grateful for the Reviewers' positive assessment of our work and thoughtful evaluation of our manuscript.

R: (major comment) The authors include a number of clustering algorithms implemented by Clusty. However, there is no detailed comparison of these clustering algorithms. In particular, how accurate are the different methods for species delineation? What are the optimal cutoffs (ANI/AF) and clustering algorithm for identifying vOTUs (MIUVIG definition and ICTV definition). More generally, how do intra-cluster and inter-cluster distances vary across each clustering method?

A: Thank you for raising this important point. Regarding the delineation of viral species, the International Committee on Taxonomy of Viruses (ICTV) only advises using the total ANI (tANI) threshold of $\geq 95\%$ for assigning two phage genomes to the same species. However, the committee does not provide any guidance on which clustering algorithm should be used to group more than two genomes into the same species. The ICTV-recommended tool, VIRIDIC, uses a complete-linkage clustering algorithm by default, which ensures that the tANI $\geq 95\%$ criterion is met for every possible pair of genomes within a cluster. For this reason, in our manuscript, we used the complete-linkage algorithm to cluster phage genomes into species and to assess the agreement of the clustering results with both VIRIDIC and the ICTV.

In response to the Reviewers' suggestion, we have further tested the performance of other Vclust clustering algorithms implemented in Clusty and compared their agreement with ICTV species clusters. All six clustering algorithms obtained a high agreement with ICTV species, but the relatively small number of genomes with established ICTV taxonomy may limit the ability to detect differences between the clustering algorithms. For instance, both single-linkage and complete-linkage clustering produced similar agreement with ICTV species taxonomy (95.9% and 95.4%, respectively), despite their differing approaches—single-linkage tends to form larger, more loosely connected clusters, while complete-linkage generates smaller, more compact clusters. Consequently, while our findings suggest that certain algorithms may perform better under specific conditions, we are cautious about recommending any particular method at this stage. Nonetheless, we have included the comparison of clustering algorithms with the ICTV species taxonomy as supplementary data

to provide further insights (**Table S5**). This table also presents a comparison of species-level clusters generated using tANI values obtained by other tools (i.e., VIRIDIC, skani, and FastANI).

With regard to the clustering of viral contigs into vOTUs, similar to ICTV, MIUViG also does not provide specific guidelines for selecting a clustering algorithm other than recommending thresholds of ANI $\geq 95\%$ and alignment fraction (AF) $\geq 85\%$. Following the Reviewers' suggestion, we compared the ANI distribution between genome pairs belonging to the same and different clusters. Among the six clustering algorithms tested, the Leiden algorithm produced clusters with relatively low content of below-threshold pairs and simultaneously left only a small number of above-threshold pairs outside as shown in new **Fig. S1**. This analysis also supports the choice of this algorithm in the latest release of the IMG/VR v4.1 database.

- R:** (major comment) Also, Figure 1c is not particularly helpful for explaining how the various clustering algorithms work.
- A:** We have revised **Fig. 1c** to show the operational principles of Vclust's clustering algorithms more clearly. Additionally, we have added **Extended Data Fig. 1**, which provides a more detailed depiction of these algorithms, extending the information presented in Fig. 1c.
- R:** (major comment) It has also been suggested that ICTV-genera for phages should be determined using an ANI threshold of $>70\%$ (10.3390/v13030506). Given that Vclust appears to perform well at lower ANI values, it would be useful to include some analyses examining its ability to cluster phage genomes at this threshold. How accurate are the different methods for genus delineation, and what are the optimal parameters?
- A:** This is a very important point. Indeed, Vclust performs well at the genus-level threshold of 70% tANI, as demonstrated by the tests using both phage genomes with simulated mutations (Fig. 2a) and genomes with natural mutations (Fig. 2b). However, evaluating ANI methods for genus delineation is challenging because the recommended 70% tANI threshold is not applied consistently across ICTV genera. For instance, in nearly 10% of the 614 ICTV phage genera ($n = 58$) containing more than one species, no interspecies genome pair within a genus meets the 70% tANI threshold (i.e., the highest tANI in the genus cluster is below 70%). Moreover, an additional 7% of these multi-species genera ($n = 43$) have a mean tANI below 70%, indicating that, at best, only a subset of genome pairs within these clusters meet the threshold. Further complicating the issue, for 9.8% of multi-species genera ($n = 60$), we found genomes in other genera that have tANI $> 70\%$. In total, we estimate that around 12,000 genome pairs require taxonomic review. This number is two orders of magnitude greater than that at the species level, presented in the first version of this manuscript (**Tables S3 and S4**). We are currently preparing a more detailed analysis of all these inaccuracies as

part of the ICTV Computational Taxonomy Challenge project (<https://ictv-vbeg.github.io/ICTV-TaxonomyChallenge/>).

Despite the above-mentioned problems, following the Reviewers' suggestion, we have evaluated Vclust's clustering accuracy at the genus level using the ICTV taxonomy as a reference. Using the same complete-linkage clustering as in the case of species-level assignment, both VIRIDIC and Vclust showed moderate agreement with the ICTV taxonomy, with VIRIDIC slightly outperforming Vclust (65% vs. 63% agreement). Using single-linkage clustering, the agreement with ICTV increased to 92% for Vclust and 93% for VIRIDIC, again highlighting that most genus clusters contain only some genome pairs that meet the recommended 70% tANI threshold. We have included these results in the revised manuscript (**lines 84-87**), and provided **Extended Data Fig. 2** illustrating the described challenge, and noted the genus-level inconsistencies in **Tables S6** and **S7**.

R: (major comment) Accuracy of Vclust should be evaluated across varying genome size, ranging from the smallest (e.g. <1 Kb) to largest (e.g. >1 Mb) viruses.

A: Thank you for this suggestion. We have evaluated Vclust and other tools on varying lengths of contig sequences from smallest (<5 kb) to largest viruses (>100 kb). The correlation of ANI between Vclust and BLASTn, across different contig sizes, ranges from 0.95 to 0.98, while the correlation of AF ranges from 0.97 to 0.99. Interestingly, the correlation between Vclust and BLASTn for the products of ANI and AF (ANI x AF) is even higher and more consistent (0.994 to 0.996). This shows that any variation in one measure (ANI or AF) between the two tools is compensated by the other, resulting in a more stable overall agreement between Vclust and BLASTn. We have added a sentence in the manuscript summarizing these results (**lines 100-101**) and detailed them in **Table S10**.

R: (major comment) Also, some tools fail to accurately calculate AF when the target genome is circularly permuted (e.g. CD-HIT) – do Vclust and other tools suffer from this same issue?

A: The Reviewers are correct that tools like CD-HIT struggle to accurately calculate AF when homologous segments between two viral genomes are reordered (e.g., due to translocations or circular permutation). This AF inaccuracy arises because CD-HIT relies on a single local alignment between two genomes to estimate ANI and AF, ignoring other homologous segments that may be rearranged, which leads to an underestimation of AF value.

In contrast, Vclust is robust to sequence rearrangements, including translocations and circularly permuted genomes. The tool identifies all local alignments between two genomes and collectively calculates ANI and AF across these alignments, even when homologous segments are out of order. To demonstrate this, we used a dataset of 4,244 complete phage genomes and its reordered version, where two regions in each genome were swapped at a random breakpoint. The comparison of AF values obtained within original datasets to their

counterparts in reordered one revealed that the mean absolute error in AF was negligible (0.04%), with a maximum observed error of 1.5%. These slight differences are due to short alignment discontinuities at the breakpoint positions in circular genomes.

In the revised manuscript, we have detailed these results in **Extended Data Fig. 5** and highlighted, in the Main Text, Vclust's capability to handle circularly permuted genomes (**line 118**).

R: (major comment) The authors should perform a more detailed scalability analysis. How do RAM and CPU scale with increasing # of genomes? How does RAM and CPU scale with relatedness of genomes (e.g. very dense vs very sparse graphs)?

A: In the revised manuscript, we have provided two experiments to show Vclust's scalability with an increasing number of genomes (**Extended Data Fig. 8a**) and increasing relatedness of genomes (**Extended Data Fig. 8b**). These tests show that both runtime and memory usage increase slightly faster than linearly with the number of sequences, while for datasets with increasing genome relatedness, runtime grows linearly with matrix density. Memory usage, however, increases more sharply for the *align* step, surpassing the *prefilter* step when genome relatedness exceeds 1% matrix density. Additionally, we conducted further tests to evaluate the effect of prefiltering parameters on scalability. Processing datasets in smaller, equally sized batches significantly reduces memory usage and slightly increases runtime without affecting results (**Extended Data Fig. 4a**). Analyzing a fraction of *k*-mers per genome (e.g., 20%) significantly reduces both runtime and memory usage, with minimal impact on sensitivity (**Extended Data Fig. 4b**). These results are detailed in the new Methods section "Optimizing performance for highly redundant genome datasets" (**lines 340-355**).

R: (major comment) How does the prefiltering step affect Vclust's speed/scalability? Is prefiltering to a `--min-ident` of 0.7 necessary for vOTU clustering (identifying all genome pairs with >95% ANI)? Or can this value be raised to decrease the pairwise comparisons performed by the align step? How accurate is the prefilter step relative to the align step?

A: The prefiltering step significantly influences the speed of Vclust as it determines the number of pairwise alignments to be performed. This can be controlled by adjusting the minimum number of common *k*-mers between two genomes (`--min-kmers`) and their minimum sequence identity (`--min-ident`). The sequence identity is calculated from *k*-mers, similar to ANI in Mash, with the exception that Vclust's calculation is relative to the shorter sequence. Consequently, the sequence identity in the prefiltering step is typically higher than the ANI derived from the alignment step. Therefore, the default `--min-ident` of 0.7 can be safely increased to approach the final alignment-based ANI threshold. For vOTU clustering (ANI \geq 95% and AF \geq 85%), increasing `--min-ident` to even 0.95 during prefiltering does

not exclude any genome pairs with an alignment-based ANI of $\geq 95\%$ (Figure below, this figure is a crop from the new **Fig. S2**).

Figure. Comparison of ANI values from the alignment step and sequence identity from the prefiltering step for 4,209,138 genome pairs meeting the MIUViG thresholds (ANI $\geq 95\%$ and AF $\geq 85\%$). The analysis was conducted on a dataset of 94,225 metagenomic viral contigs sampled from IMG/VR v4.1, as used in Fig. 2d.

The effect of increasing `--min-ident` on the number of genome pairs requiring alignment is shown in Table below. For instance, raising `--min-ident` from 0.7 to 0.95 reduces the number of genome pairs to be aligned by fourfold, while still recalling all pairs with alignment-based ANI $\geq 95\%$.

Table. Effect of `--min-ident` on the number of genome pairs passing the prefiltering step and the number of recalled genome pairs with ANI $\geq 95\%$ and AF $\geq 85\%$.

Minimum sequence identity (prefilter)	Number of genome pairs passing prefilter	Number of recalled genome pairs with ANI $\geq 95\%$ and AF $\geq 85\%$ (align)
0.70	56,549,952	4,209,138 (100%)
0.90	25,106,519	4,209,138 (100%)
0.91	22,321,019	4,209,138 (100%)

0.92	19,703,794	4,209,138 (100%)
0.93	17,113,368	4,209,138 (100%)
0.94	14,705,814	4,209,138 (100%)
0.95	12,467,035	4,209,138 (100%)

In the revised manuscript, we have described the relationship between the prefilter's sequence identity and the final alignment-based ANI in the Methods section (**lines 316-324, Fig. S2**) and Vclust's GitHub documentation. Additionally, the documentation now includes examples of using `--min-ident` values other than the default 0.7, such as for assigning viral contigs to vOTUs using the MIUViG thresholds.

R: (major comment) Can the authors please provide data supporting their choice in default parameters? Anchor length, seed length, window length, etc. should be included.

A: Of course. Our primary objective in developing Vclust was to ensure that its alignments were at least as accurate as those generated by BLASTn, which is considered the most reliable tool for estimating ANI. BLASTn uses a default word size of 11 nucleotides (i.e., the length of the initial match), in contrast to less sensitive MegaBLAST, which uses a larger word size of 28 nucleotides. Consequently, we set the anchor length in Vclust to 11 nucleotides to align with the parameters used by BLASTn.

The remaining parameter values (e.g., seed length, window length) were selected through Bayesian optimization using Gaussian process minimization. This optimization was carried out on a dataset of 10,000 viral genome pairs, each with simulated mutations (i.e., various combinations of substitutions, deletions, insertions, inversions, duplications, and translocations) and known tANI value in the range of $\geq 70\%$. The optimization aimed to minimize the mean absolute error (MAE) between the predicted tANI values and the known reference tANI values, with parameters yielding the lowest MAE selected as defaults.

In the revised manuscript, we have provided the rationale for selecting the default parameters in the new Methods section "Alignment parameters" (**lines 255-264**). Additionally, we have included the detailed results of the Bayesian optimization process in **Table S13**.

R: (major comment) The performance of Vclust should be benchmarked vs other tools using different #s of CPUs (e.g. 1, 2, 4, 8, 16, 32). In Figure 2c Vclust is compared to other tools using 32 CPUs, indicating that MEGABLAST would take >4 years of CPU time to process IMG/VR v4. It is known that tools like MegaBLAST scale poorly with the number of CPUs.

A: In the revised manuscript, we have provided a detailed comparison of Vclust's performance against other tools across a range of CPU threads (1 to 64) in **Extended Data Fig. 3**. Due to the long execution time of MegaBLAST, the analysis was conducted on a subset of 1 million contigs sampled from the IMG/VR database. The results indicate that while most tools

exhibit a linear decrease in runtime up to 8 threads, some—including Vclust—show reduced speed-up beyond this point. Vclust has a minor slowdown with 64 threads compared to 32, due to task fragmentation overhead, but it still ranks as the second fastest tool, just behind MMseqs2. Importantly, when applied to the entire IMG/VR database of 15 million viral contigs, Vclust outperforms MMseqs2 in both speed and memory efficiency (**Fig. 2e**), demonstrating its scalability for very large datasets.

- R:** (minor comment) The same set of tools should be evaluated in all benchmarks. CheckV's ANIcalc is left out of initial benchmarks but included in latter ones. Please report performance in the text and in figure a (panels a, b, & c)
- A:** We understand the importance of consistency in benchmarking across tools. However, due to differences in supported similarity measures (e.g., tANI vs. ANI), some tools in our study are not directly comparable.

In our initial benchmarks, we evaluated the tANI measure recommended by the ICTV's Bacterial Viruses Subcommittee for species and genus delineation in complete bacteriophage genomes. VIRIDIC, specifically designed for this task, calculates tANI through highly sensitive BLASTn-based searches and detailed analysis of local alignments (HSPs), eliminating the risk of double-counting matched or mismatched nucleotides across overlapping HSPs. Although computationally demanding, VIRIDIC's methodology ensures accurate tANI calculations, making it a suitable reference for benchmarks emphasizing precision. In contrast, CheckV's anicalc does not calculate tANI but estimates ANI and AF based on user-provided BLAST results. Due to its reliance on less sensitive BLAST searches and a weighted average of percent identities from HSPs rather than an exact count of matched and mismatched nucleotides, anicalc is inherently less accurate than VIRIDIC. For this reason, we prioritized VIRIDIC in our tANI-focused benchmarks due to its precision and its relevance as Vclust's main competitor. Furthermore, our choice is also influenced by the size constraints imposed by the journal on *Brief Communication* articles. To ensure a comprehensive assessment, we included skani and FastANI, which represent alignment-free and semi-alignment-based methods, respectively, and are primary competitors of Vclust in computational performance. We believe this benchmarking strategy is justified, as it allows us to compare the most accurate tools when evaluating accuracy and the fastest tools when assessing computational efficiency.

- R:** (minor comment) The authors claim that Kmer-db2 outperforms Kmer-db1, but no data is given to support this claim.
- A:** In the revised manuscript, we have conducted a performance comparison between Kmer-db 1 and Kmer-db 2 using a random subsample of 10k, 20k, 50k, 100k, 200k, 500k, and 1M IMG/VR contig sequences. As shown in **Extended Data Fig. 6a**, Kmer-db 2 is ~3x faster than Kmer-db 1 during database construction (*build* mode), an advantage that remains stable

across the tested sequence ranges. In all-to-all sequence comparisons, the sparse mode introduced in Kmer-db 2 (*all2all-sp*) significantly outperformed Kmer-db 1. For example, in the case of the 500k sequence dataset, it was over 100x faster than original *all2all* mode (**Extended Data Fig. 6b**). Additionally, memory consumption in the standard *all2all* mode becomes a limiting factor, with computations only feasible up to 500k sequences due to the dense matrix requirements. In contrast, Kmer-db 2 maintains significantly lower RAM usage, enabling efficient handling of much larger datasets. In the revised manuscript, this analysis is detailed in **Extended Data Fig. 6** and referenced in the Methods section (**lines 207-209**).

R: (minor comment) [Line 101-102] Language / typo

A: Thank you for pointing out this unfortunate language. We have revised the sentence (**lines 109-110**).

R: (minor comment) [Figure 2a] What parameters were used for VIRIDIC? The VIRIDIC commands for 2b and 2c are given in the supplementary tables, but this is not given for Fig 2a.

A: We apologize for the oversight. We have now included the parameters for VIRIDIC and other tools in the Methods section (**lines 363-367**).

R: (minor comment) [Table S11] “Notably, vOTU clusters obtained from Vclust and MegaBLAST exhibited high adjusted rand scores”. This logic is circular since IMG/VR vOTUs were generated using MegaBLAST + Leiden clustering.

A: We focused the manuscript primarily on comparing ANI values between Vclust and MegaBLAST/BLASTn-based tools, as these are recognized as the most accurate methods for ANI estimation. We believe that the sentence in the Table S11 caption, highlighted by the Reviewers, informs readers that Vclust also – in addition to ANI values – agrees with MegaBLAST in clustering results across different clustering algorithms. We have revised the sentence for clarity.

R: (minor comment) [Github] Tool documentation and setup is fairly straightforward, but creating a BioConda package would greatly increase this tool’s accessibility and usage.

A: Great suggestion, thank you. Vclust is now available through Bioconda and PyPI, with updated GitHub documentation.

R: (minor comment) [Vclust CLI] An option to output the alignment coordinates in addition to the number of alignments could allow more detailed comparison of the differences between Vclust and alternative alignment tools.

A: We have added an option to Vclust to output alignment details in a separate TSV file. This file is similar to the BLASTn/MegaBLAST tabular output and includes detailed information

on each local alignment between two genomes, such as the coordinates in both the query and reference sequences, strand orientation, the number of matched and mismatched nucleotides, and the percentage of sequence identity. This new feature is described in the revised manuscript in the Methods section (**lines 330-332**) and has also been documented in our GitHub repository, with Vclust's version updated from v1.0 to v1.2 to reflect these changes.

- R:** (minor comment) An exploration of alignments identified by Vclust that were missed by alternative alignment tools would be very informative (how many of these regions are there? what is the ANI of these regions? Why might these regions be missed by megaBLAST/blastn?)
- A:** We have examined local alignments generated by Vclust, MegaBLAST, and BLASTn. Briefly, Vclust shows a high level of agreement with BLASTn at the nucleotide level, with 96% of nucleotides commonly aligned by both tools. On average, Vclust generates more local alignments per genome pair (mean = 65) compared to BLASTn (mean = 53) and MegaBLAST (mean = 25). Although Vclust's alignments are shorter in length than BLASTn (mean 11,644 bp versus BLASTn's 19,008 bp), they have comparable average ANI (85%-86%). The results of this analysis are now included in **Extended Data Fig. 7** and are referenced in the Methods section (**lines 264-266**).